# Workplace Integrated Safety and Health Program Uptake in Nursing Homes: Associations with Ownership

**DOI:** 10.3390/ijerph182111313

**Published:** 2021-10-28

**Authors:** Jamie E. Collins, Leslie I. Boden, Daniel A. Gundersen, Jeffrey N. Katz, Gregory R. Wagner, Glorian Sorensen, Jessica A. R. Williams

**Affiliations:** 1Brigham and Women’s Hospital, Boston, MA 02115, USA; jnkatz@bwh.harvard.edu; 2Harvard Medical School, Boston, MA 02115, USA; 3School of Public Health, Boston University, Boston, MA 02118, USA; lboden@bu.edu; 4Dana-Farber Cancer Institute, Boston, MA 02115, USA; daniela_gundersen@dfci.harvard.edu (D.A.G.); Glorian_Sorensen@dfci.harvard.edu (G.S.); 5Harvard T.H. Chan School of Public Health, Boston, MA 02115, USA; gregory_wagner@comcast.net; 6College of Health and Human Development, The Pennsylvania State University, State College, PA 16801, USA; jqw6242@psu.edu

**Keywords:** nursing homes, occupational health, total worker health

## Abstract

Workers in nursing homes are at high risk of occupational injury. Understanding whether—and which—nursing homes implement integrated policies to protect and promote worker health is crucial. We surveyed Directors of Nursing (DON) at nursing homes in three US states with the Workplace Integrated Safety and Health (WISH) assessment, a recently developed and validated instrument that assesses workplace policies, programs, and practices that affect worker safety, health, and wellbeing. We hypothesized that corporate and for-profit nursing homes would be less likely to report policies consistent with Total Worker Health (TWH) approaches. For each of the five validated WISH domains, we assessed the association between being in the lowest quartile of WISH score and ownership status using multivariable logistic regression. Our sample included 543 nursing homes, 83% which were corporate owned and 77% which were for-profit. On average, DONs reported a high implementation of TWH policies, as measured by the WISH. We did not find an association between either corporate ownership or for-profit status and WISH score for any WISH domain. Results were consistent across numerous sensitivity analyses. For-profit status and corporate ownership status do not identify nursing homes that may benefit from additional TWH approaches.

## 1. Introduction

The Total Worker Health (TWH) Initiative was launched by the National Institute for Occupational Health and Safety (NIOSH) in 2011 as a strategy to integrate occupational safety, health protection, and health promotion, with the ultimate goal to protect and promote worker wellbeing, health, and safety [1,2,3]. TWH shows promise in improving the health, safety, and wellbeing of workers [4,5]. Research on the relationship between TWH principles and worker and enterprise outcomes must consider the degree to which TWH policies and practices are implemented in specific workplaces.

The Workplace Integrated Safety and Health Assessment (WISH) is an organizational level measure of workplace policies, programs, and practices that affect worker safety, health and wellbeing [6]. The tool, informed by the Total Worker Health^®^ framework, was developed to allow employers and researchers to assess how well the best practices for supporting work safety, health, and wellbeing are implemented and was recently validated using item response theory analysis [7]. There are five domains in the validated measure: leadership commitment; participation; policies, programs, and practices that foster supportive working conditions; comprehensive and collaborative strategies; and adherence to federal and state regulations and ethical norms. Administration of the WISH will allow for the evaluation of various hypotheses regarding TWH policies and practices and associated outcomes; for example, do institutions with high scores on the Policies, Programs, and Practices domain have lower rates of worker injury? Previous work provides a comprehensive overview of the WISH, including the conceptual model describing the central role of working conditions in shaping health and safety outcomes and how each domain reflects these conditions [6].

The implementation of TWH principles may be especially important for workplaces with low-wage workers where other approaches to addressing wellbeing, such as individual-centered wellness programs, have been ineffective [8,9]. In this analysis, we focus on long-term care facilities (nursing homes) because their worker population has relatively low wages and high rates of non-fatal work-related injury (5.9 per 100 full-time workers in 2019) [10,11,12]. The nursing home industry disproportionately employs groups identified by NIOSH as at-risk for occupational health disparities [13,14]. Nursing home employees can face both physically demanding job tasks and stressful work environments, leading to increased risks of physical health problems, such as musculoskeletal disorders and cardiovascular disease, as well as mental health problems [15,16,17,18]. Therefore, health promotion is especially important in this population. Additionally, there are over 15,000 nursing facilities in the U.S. that are certified to provide care and receive payment from the Centers for Medicare and Medicaid Services (CMS), the primary payer for long-term care services in the U.S. [19]. 

Nursing home ownership influences the operational philosophy and priorities of nursing homes [20]. These differences may also be reflected in the nursing home’s approach to its workforce, as measured by the WISH. However, the literature is conflicted regarding the relationship between the ownership status of U.S. nursing homes and quality of care, with some studies suggesting that for-profit or chain status may influence quality of care [21,22] and others suggesting that ownership and management structures may be more nuanced than a simple for-profit vs. not for-profit designation [23,24]. We were unable to find any published manuscripts that have investigated whether ownership type influences the implementation of TWH policies and practices.

The purpose of this study is to determine the degree to which nursing homes implement TWH approaches as measured by the WISH and determine whether their ownership type, for-profit and/or corporate, is associated with their scores, after controlling for other organizational characteristics. We hypothesize that, after adjusting for nursing home characteristics, (1) for-profit ownership will be associated with a lower uptake of THW approaches, as measured by the WISH, (2) corporate ownership will be associated with a lower uptake of THW approaches, as measured by the WISH. To our knowledge, this is the first comprehensive survey in nursing homes of the implementation of workplace policies, programs, and practices that can affect worker safety, health, and wellbeing.

## 2. Materials and Methods

Study Design. We conducted a cross-sectional survey of all operating nursing homes that were certified by CMS in California, Massachusetts, and Ohio. The survey took about 20 min to complete and covered a wide array of topics relating to the working environment of the nursing home. The survey was subject to cognitive testing to ensure face validity [7]. We merged these data with those available from the Certification and Survey Provider Enhanced Reporting system to match each nursing home that answered the survey to relevant organizational characteristics. The study was approved by the Institutional Review Board of the Harvard T.H. Chan School of Public Health (IRB 18-1245).

Setting and Participants. The survey was conducted in two waves between October 2018 and June 2019. All nursing homes that served adults, had at least 30 beds, were open, and were certified by CMS were included in the study. The self-administered survey was initially sent out electronically via email link to the Directors of Nursing (DON) of each facility. There were three electronic follow-ups with the final follow-up, sent as a paper survey to the DON. DONs were instructed to seek input or pass the survey to an appropriate health and safety representative for their nursing home, if needed. Approximately 35% of the sample also received a follow-up reminder phone call (randomly selected). Upon completion of the survey, respondents were sent Amazon gift cards. The overall response rate for the survey was 23.8% (569/2389). In previous work, we did not find any relationship between organizational characteristics, quality ratings, or health inspection citations with whether a nursing home responded to the survey [25].

Variables. The WISH was measured using the survey. The WISH instrument includes closed-ended questions, each with one of two four-point ordinal response scales ((0) not at all, (1) somewhat, (2) mostly, (3) completely; (0) not at all, (1) some of the time, (2) most of the time, (3) all of the time). Policies, Programs, and Practices contains 10 items with a total score ranging from 0 to 33. The other domains each contain four items and total scores ranging from 0 to 12. Higher scores reflect greater implementation of TWH principles. All domains exhibited scores skewed toward higher (more favorable) scores, therefore, for statistical modeling, we dichotomized each domain as the bottom 25th percent vs. top 75th percent. This cut-point was based on the data distribution; in sensitivity analyses, we dichotomized each domain at approximately the 10th percentile (bottom 10% vs. top 90%) and at approximately the 33rd percentile (bottom one-third vs. top two-thirds). As a final sensitivity analysis, we compared those nursing homes scoring in the bottom 25th percent to those in the top 25th percent. 

The key predictor was ownership status according to Medicare provider files. Medicare data on federally certified nursing homes come from state surveys in each facility, which must occur at least once every 15 months. Publicly available data from the provider files define ownership according to the following categories: For-Profit (Corporation, Individual, LLC, or Partnership), Government (city, city/county, county, federal, hospital district, or state), or Non-Profit (church-related, corporation, or other). Given that we did not have government facilities in our sample, and based on the distribution of the sample across the remaining categories, we defined ownership in two ways, as for-profit or not-for-profit and corporate vs. non-corporate. Models were run separately for each ownership variable. As a secondary analysis, we considered four-level ownership: (1) corporate, for-profit, (2) corporate, not-for-profit, (3) non-corporate, for-profit, (4) non-corporate, not-for-profit.

Covariates included the number of federally certified beds, occupancy rate, percent of residents who are Medicaid recipients, and staffing patterns (RN ratio, LPN ratio, and CNAs ratio) according to Centers for Medicare and Medicaid Services. Nursing home location was classified as rural vs. non-rural using the most recent (2010) Rural–Urban Commuting area codes [26]. Codes 1–3 indicate metropolitan areas; all other codes were considered rural. By including all of these covariates we are estimating the marginal association of ownership type with the WISH. 

Additionally, we included indicator variables for each state (using one as a reference), since each state has differences in how nursing homes are regulated and differ in many other ways. As mentioned above, the survey was operated as two separate waves, so we also included an indicator for whether the survey was in the first or second wave. 

Statistical Methods. We present descriptive statistics for each WISH domain. We assessed the correlation between domains scores using Spearman correlation. We used multivariable logistic regression models to assess the association between dichotomous WISH score and ownership, with errors clustered at the state level. We report the results using odds ratios and simultaneous 95% Confidence Intervals. We added covariates to the model in three stages. Model 1 included only the survey wave and state. Model 2 additionally adjusted for all covariates with the exception of staffing patterns. We were concerned that staffing patterns may be mediators—of the causal pathway between ownership and WISH scores—thus, we included staffing patterns in a final exploratory Model 3.

Our initial power calculations assumed that the WISH domains would be analyzed as continuous outcomes. We anticipated obtaining surveys from 200 for-profit and 200 not-for-profit nursing homes, which would provide >80% power to detect an effect size of approximately 0.3 standard deviations. Recalculating power, assuming a dichotomous outcome with approximately 25% prevalence, finds a detectable OR of approximately 1.75.

## 3. Results

We received surveys from 569 out of 2389 nursing homes. A detailed description of the survey methods and predictors of survey response is provided in a previous work, which did not find any statistically significant predictors of response [25]. Two surveys did not include sufficient detail to compute any WISH domain; an additional 24 nursing homes were missing at least one covariate. These nursing homes were excluded from the analysis, leaving 543 in the final analytic sample.

Descriptive statistics of the nursing home characteristics are presented in Table 1. Seventy-seven percent of the nursing homes had for-profit ownership and 83% were owned by corporations. The average number of beds was 101. Thirty-seven percent of nursing homes were in California, 22% in Massachusetts, and 41% in Ohio.

### 3.1. Reported Implementation of TWH Approaches

The reported implementation of TWH approaches was high. Descriptive statistics for each WISH domain are shown in Table 2 and histograms are shown in Appendix A. The median score for the Policies, Programs, and Practices was 28 (maximum possible 33), with 12% of respondents at the maximum possible score. The median scores for the other four domains (maximum possible score 12) ranged from 9 to 12. The Adherence domain had the highest average score (10.7 out of 12), while the Participation domain had the lowest average score (8.8 out of 12). Each domain score was dichotomized at the 25th percentile, or the bottom of the IQR (e.g., adherence was categorized as <10 vs. 10, 11, 12). We chose to dichotomize the measure to see whether the hypothesized characteristics were associated with low performers. These low performers would be a high priority for interventions to improve the working environment. We tested a few different cut points (such as the 10th percentile) with similar results.

The correlation between WISH domains was moderate to high, ranging from 0.55 (Participation vs. Adherence) to 0.78 (Policies, Programs, and Practices vs. Comprehensive and Collaborative Strategies) (Appendix A). Of the 543 nursing homes included in this analysis, 268 (49%) were in the top 75% on all five domains; 39 (7%) were in the bottom quartile on all five domains.

### 3.2. Associations between Nursing Home Owership and TWH Approaches

Appendix A display the association between each continuous WISH domain score and ownership in boxplots (2A: not-for-profit vs. for-profit, 2B: non-corporate vs. corporate). For both ownership variables, the distribution of each continuous WISH domain score is similar between ownership categories.

The adjusted odds of being in the bottom quartile of each WISH domain (reporting the least implementation of TWH program elements) for nursing homes with for-profit vs. not-for-profit ownership are shown in Table 3. We observed no significant associations between for-profit ownership and any WISH domain. There were trends showing slightly increased odds of for-profit nursing homes in the bottom quartile of Leadership, with for-profit nursing homes associated with 1.35 times increased odds of being in the bottom Leadership quartile. There were also slightly decreased odds of being in the bottom quartile of Participation and Comprehensive and Collaborative Strategies, but these associations were modest and not statistically significant. The results were similar in unadjusted analyses, and in analyses that were additionally adjusted for staffing patterns (Appendix A).

The adjusted odds of being in the bottom quartile of each WISH domain for nursing homes with corporate vs. non-corporate ownership are shown in Table 4. We observed no significant associations between for-profit ownership and any WISH domain. Results were similar in unadjusted analyses, and in analyses additionally adjusted for staffing patterns (Appendix A).

Sensitivity analyses investigating a combined four-level ownership status were largely in line with the primary analyses (Table 5). Compared to non-corporate, not-for-profit nursing homes, all other ownership categories were at slightly decreased odds of being in the bottom quartile for Participation and Comprehensive and Collaborative Strategies, and slightly increased odds of being in the bottom quartile for Leadership. As with the primary analyses, none of the associations were statistically significant.

Results from the sensitivity analyses investigating different cut-points for dichotomizing each WISH domain were similar to the primary analyses; we found no statistically significant associations between any WISH domain and either ownership variable [Appendix A]. Finally, sensitivity analyses comparing those homes scoring in the bottom 25th percentile to those in the top 25th percentile were similar to the primary analysis comparing the bottom 25th percentile to the top 75th percentile (Appendix A). 

## 4. Discussion

### 4.1. Summary

In our survey of 543 DONs in three states, we found that nursing homes report a high degree of implementation of TWH approaches as measured by WISH scores. The Adherence domain, assessing “the degree to which the organization adheres to federal and state regulations, as well as ethical norms, that advance worker safety, health and well-being”, had, on average, the highest score, while the Participation domain, assessing “whether at every level of an organization, including labor unions or other worker organizations if present, help plan and carry out efforts to protect and promote worker safety and health”, had, on average, the lowest score.

### 4.2. Practical Implications

We hypothesized that nursing home ownership—for-profit vs. not-for-profit and corporate vs. non-corporate—would be associated with the implementation of TWH approaches. We did not find any statistically significant or clinically meaningful associations between either ownership definition and any WISH domain. While both for-profit and corporate-owned nursing homes had slightly increased odds of lower Leadership domain scores, they also had a slightly increased odds of having higher Participation domain scores. Thus, we did not observe a meaningful association between ownership status and implementing TWH approaches.

The topic of ownership in U.S. nursing homes, particularly whether it impacts quality of care, has a long and fraught history [23]. Many papers have compared quality based on board ownership structure and have generally found that for-profit nursing homes provide lower-quality care in several key domains. However, many of the studies suffer from methodological difficulties that make causal inference challenging [21]. Several mechanisms for this difference have been highlighted in the literature, such as the context, principal-agent issues arising from owner-managers, financial performance, and profit taking [22,27,28,29]. The comparison generally made in the U.S., between non-profit and for-profit nursing homes differs from many other countries, such as Sweden, where private nursing homes are often compared to public ones or England, where public nursing homes are compared to private for-profit and private not-for-profit nursing homes [30,31].

Additionally, the work environment as measured by the WISH does not appear to be associated with broad ownership classification as defined in this paper. Nearly 70% (in 2016) of U.S. nursing homes are for-profit; thus, most older adults are cared for in for-profit settings [32]. Comparing these for-profit settings to nonprofit settings may not be the most useful comparison, despite its prevalence in the literature, particularly considering the increasing complexity of the ownership structures of nursing homes. Previous work in nursing homes did use ownership as a shorthand for trying to measure differences in incentives and motivations among nursing home operators that might have downstream effects on the quality of care. However, researchers who hope to incorporate measures of the work environment beyond average staffing levels need to measure this separately, rather than assume that these differences between nursing homes match up with ownership.

One recent study used a 43-item tool to assess readiness for integration in six nursing homes owned by a single for-profit chain owner. This study found low variability in scores, which is unsurprising because chain ownership necessitates uniformity in programs, policies, and practice [33]. In our much larger study of over 500 nursing homes, we found relatively low variability in WISH domain scores. The bottom 25th percent represented anywhere from ~75% (Adherence, the bottom 25th percent included scores ranging from 0 to 9 out of 12) to ~50% (Participation, the bottom 25th percent included scores ranging from 0 to 6 out of 12) of the scale range. The WISH tool may, therefore, be important when identifying nursing homes with the poorest TWH policies. Identifying these homes and their areas of weakness is the first step in developing interventions to improve the work environment. 

### 4.3. Theoretical Implications

The WISH domains were developed through a comprehensive approach that includes content review by subject matter experts, cognitive testing with respondents to ensure comprehension and response mapping, and item response theory analysis. Data-driven change was included as a sixth domain, but no reasonably fitting model could be identified for the dimension in validation analysis, and thus it is not presented further [7]. Common among all domains was the observation of low information (i.e., poorer measurement precision) at the higher end of the scores. Thus, the items have limited utility when differentiating among the higher-scoring nursing homes. However, it has great utility when identifying the lower-scoring nursing homes that are priorities for intervention as we did in the present analysis. This finding is reflected in our sensitivity analyses, which showed similar results with respect to the association between ownership and WISH scores when we compared those nursing homes in the bottom quartile of WISH scores to the top three quartiles (Table 2 and Table 3) vs. those in the bottom vs. top quartile (Appendix A). Future studies should extend our analysis to examine the structural relationships between nursing home operations and WISH domains, as well as the effectiveness of interventions in producing improvements in WISH scores among nursing homes that are priorities for intervention. Such analyses should use other scoring approaches, such as modal a priori scores or fitting structural equation models that capture the granularity in the range in which the WISH tool has its greatest ability to differentiate among nursing homes, and thus should be sensitive to differences among nursing homes and/or changes following intervention. 

### 4.4. Future Directions

While the majority of surveyed nursing homes reported a high uptake of TWH policies, organizations may be interested in how they can improve the implementation of these policies and practices. The Center for Work, Health, and Well-Being at the Harvard T.H. Chan School of Public Health designed a suite of resources including Implementation Guidelines, training, and technical assistance to support the implementation of TWH policies [34]. A pilot study in three small-to-medium (less than 750 employees) organizations found that these resources were acceptable to the participating companies and could feasibly be implemented [35]. Implementing TWH interventions may be especially challenging in low-wage, high-attrition industries, such as long-term care facilities or food service [36]. A recent study highlighted how these Implementation Guidelines were utilized to develop and implement an organizational intervention to improve worker health among low-wage food service workers [37]. This work found the Implementation Guidelines to be transferable across industries. Future work should investigate whether and how such an approach could be implemented to improve worker health in long-term care facilities.

### 4.5. Limitations

This study has several limitations. The sample only included nursing homes from three US states and excluded nursing homes with fewer than 30 beds. Generalizing these findings should be done with caution; the WISH domains should be evaluated in different geographic locations and settings. The response rate of 23.8% was lower than has been cited in prior work surveying DONs [38,39]. However, a prior analysis on this sample did not find significant associations between nursing home characteristics and response [25]. The study was initially designed to assess the WISH domains as continuous scores. Dichotomizing continuous outcomes is known to reduce statistical power; however, given the highly skewed nature of the domain scores, we felt this was necessary [40]. Sensitivity analyses investigating different cut-points for dichotomization were similar to the primary analysis. Additionally, we used a relatively simple description of the ownership of nursing homes. Current ownership structures have become more complicated over time, and our measures may not reflect the differences in incentives and motivations that result from more complicated structures. Finally, all surveys were self-administered and may be subject to desirability or optimism biases. The WISH tool was designed to be completed at the organizational level by employer representatives [6]. Unlike other tools, such as the Integrated Health & Safety Index (IHS Index) and the HERO Scorecard, it was not designed to use individual employee data [41,42]. Therefore, the WISH tool could be utilized in future studies to complement surveys directed to workers in order to identify discordances between organizational policies, practices and programmers, and workers’ perceptions of these.

## 5. Conclusions

In our survey of over 500 nursing homes in three US states, we found a high degree of nursing homes that reported implementing TWH approaches as measured by the WISH. We did not find associations between for-profit vs. not-for-profit or corporate vs. non-corporate, and WISH domain scores. Simply relying on ownership status is not sufficient for researchers and policy makers seeking to identify nursing homes that may benefit from the additional implementation of TWH approaches. 

## Figures and Tables

**Table 1 ijerph-18-11313-t001:** Descriptive Statistics of Nursing Home Characteristics.

Variable	Mean (SD) or *n*	Median; Min–Max or %
Ownership		
corporate, for-profit	315	58%
corporate, not-for-profit	97	18%
non-corporate, for-profit	98	18%
non-corporate, not-for-profit	33	6%
Number of beds	101 (46)	99; 30–378
Percent Medicaid ^1^	61% (21%)	66%; 0–100%
Percent occupied ^1^	85% (12%)	89%; 23–100%
RN ratio	0.61 (0.31)	0.55; 0.04–2.54
LPN ratio	0.98 (0.36)	0.94; 0.11–3.93
CNAs ratio	2.29 (0.55)	2.25; 1.02–7.84
Location-rural	82	15%
State		
California	203	37%
Massachusetts	117	22%
Ohio	223	41%

^1^ Values over 100% truncated to 100%.

**Table 2 ijerph-18-11313-t002:** Descriptive Statistics of WISH Domains.

WISH Domain	Mean (SD)	Median	IQR
Adherence	10.7 (2.0)	12	10, 12
Comprehensive and Collaborative Strategies	9.4 (2.8)	10	8, 12
Leadership	9.5 (2.5)	10	8, 12
Participation	8.8 (2.7)	9	7, 11
Policies, Programs, and Practices	26.2 (6.0)	28	23, 31

**Table 3 ijerph-18-11313-t003:** Adjusted association between for-profit nursing home ownership and being in the bottom 25% percentile of WISH domain.

Variable	Odds Ratio ^1^	95% CI ^2^	*p*-Value
Adherence	0.98	[0.54, 1.77]	0.9364
Comprehensive and Collaborative Strategies	0.71	[0.39, 1.29]	0.2612
Leadership	1.35	[0.73, 2.50]	0.3394
Participation	0.79	[0.47, 1.34]	0.3784
Policies, Programs, and Practices	0.96	[0.53, 1.73]	0.8858

^1^ Odds ratio of being in bottom 25th percent for for-profit vs. not-for-profit nursing homes, adjusted for survey wave, state, number of beds, occupancy rate, percent of residents who are Medicaid recipients. ^2^ CI = Confidence Interval.

**Table 4 ijerph-18-11313-t004:** Association between corporate nursing home ownership and being in the bottom 25% percentile of WISH domain.

Variable	Odds Ratio ^1^	95% CI ^2^	*p*-Value
Adherence	1.23	[0.73, 2.06]	0.4412
Comprehensive and Collaborative Strategies	1.10	[0.65, 1.87]	0.7109
Leadership	1.26	[0.73, 2.16]	0.4037
Participation	0.73	[0.46, 1.16]	0.1887
Policies, Programs, and Practices	1.00	[0.6, 1.67]	0.9911

^1^ Odds ratio of being in bottom 25th percent for corporate vs. non corporate nursing homes, adjusted for survey wave, state, number of beds, occupancy rate, percent of residents who are Medicaid recipients. ^2^ CI = Confidence Interval.

**Table 5 ijerph-18-11313-t005:** Association between corporate nursing home ownership and being in the bottom 25% percentile of WISH domain.

Variable	Odds Ratio ^1^	95% CI ^2^	*p*-Value
Adherence			0.8879
corporate, for-profit	1.24	[0.47, 3.32]	
corporate, not-for-profit	1.33	[0.46, 3.81]	
non-corporate, for-profit	1.04	[0.35, 3.06]	
non-corporate, not-for-profit	Reference		
Comprehensive and Collaborative Strategies			0.5643
corporate, for-profit	0.64	[0.25, 1.60]	
corporate, not-for-profit	0.80	[0.29, 2.18]	
non-corporate, for-profit	0.50	[0.18, 1.41]	
non-corporate, not-for-profit	Reference		
Leadership			0.6633
corporate, for-profit	1.56	[0.55, 4.42]	
corporate, not-for-profit	1.13	[0.36, 3.52]	
non-corporate, for-profit	1.22	[0.39, 3.83]	
non-corporate, not-for-profit	Reference		
Participation			0.4935
corporate, for-profit	0.59	[0.26, 1.35]	
corporate, not-for-profit	0.73	[0.30, 1.78]	
non-corporate, for-profit	0.79	[0.32, 1.97]	
non-corporate, not-for-profit	Reference		
Policies, Programs, and Practices			0.9992
corporate, for-profit	0.97	[0.38, 2.49]	
corporate, not-for-profit	1.01	[0.36, 2.84]	
non-corporate, for-profit	0.96	[0.34, 2.72]	
non-corporate, not-for-profit	Reference		

^1^ Odds ratio of being in bottom 25th percent of each WISH domain compared to non-corporate, not-for-profit nursing homes, adjusted for survey wave, state, number of beds, occupancy rate, percent of residents who are Medicaid recipients. ^2^ CI = Confidence Interval.

## Data Availability

Data available on request due to privacy restrictions. The data presented in this study are available on request from the corresponding author. The data are not publicly available to reduce the risk of identification of the subjects who are often the only person with their job title at an individual nursing home.

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
