# Peer review of "Workplace Integrated Safety and Health Program Uptake in Nursing Homes: Associations with Ownership"

_ijerph, 2021, doi:10.3390/ijerph182111313_

Round 1

Reviewer 1 Report

This study performs an assessment of the Total Worker Health (TWH), which is a strategy to integrate occupational safety, health protection, and health promotion, at nursing homes in the US. The magnitude of implementation of TWH was measured with the Workplace Integrated Safety and Health Assessment (WISH), an organizational level measure of workplace polices, programs, and practices that affect worker safety, health and wellbeing. In particular, directors of nursing at nursing homes in a total of three States in the US were surveyed to collect data to determine the WISH. Moreover, the impact of nursing homes’ ownership type on the WISH, and thereby, implementation of TWH was analyzed.

This is a well-written manuscript with an accurate research design. It demonstrates the level of novelty required for publication. The manuscript includes a few flaws. A review of relevant research works should be presented. Moreover, there is a lack of whitespace between text and reference. For example, “worker wellbeing, health, and safety[1-3].” should be “worker wellbeing, health, and safety [1-3]. In addition, it is advised to avoid the word “robust” in this study because it has different meanings in different disciplines.

Reviewer 2 Report

Comment on IJERPH

  1. The major concern to the manuscript is the relationships that you intend to build and test. I would suggest your having specific relationships in the theoretical part. I help build some hypotheses for your references.

Hypothesis 1:

Nursing home adherence is positively related to worker safety (H1a), health (H1b) and wellbeing (H1c).

Hypothesis 2:

Collaborative strategy is positively related to worker safety (H2a), health (H2b) and wellbeing (H2c).

Hypothesis 3:

(Servant) leadership is positively related to worker safety (H3a), health (H3b) and wellbeing (H3c).

Hypothesis 4:

Participation in decision-making is positively related to worker safety (H4a), health (H4b) and wellbeing (H4c).

Hypothesis 5:

Employee-oriented work system is positively related to worker safety (H5a), health (H5b) and wellbeing (H5c).

Hypothesis 6:

Ownership moderates the relationship between nursing home adherence and worker safety (H6a), health (H6b) and wellbeing (H6c) such that their positive relationship is stronger for non-for-profit nursing homes than for-profit nursing homes.

Hypothesis 7:

Ownership moderates the relationship between nursing home collaborative strategy and worker safety (H7a), health (H7b) and wellbeing (H7c) such that their positive relationship is stronger for non-for-profit nursing homes than for-profit nursing homes.

Hypothesis 8:

Ownership moderates the relationship between nursing home collaborative strategy and worker safety (H8a), health (H8b) and wellbeing (H8c) such that their positive relationship is stronger for non-for-profit nursing homes than for-profit nursing homes.

Hypothesis 9:

Ownership moderates the relationship between nursing home servant leadership and worker safety (H9a), health (H9b) and wellbeing (H9c) such that their positive relationship is stronger for non-for-profit nursing homes than for-profit nursing homes.

Hypothesis 10:

Ownership moderates the relationship between nursing home employee-oriented work system and worker safety (H10a), health (H10b) and wellbeing (H10c) such that their positive relationship is stronger for non-for-profit nursing homes than for-profit nursing homes.

  1. Rewrite your introduction. Go straight forward your research motivation. What’s the research problem in the current research on the impact of contextual factors on worker safety, health and wellbeing. How will your study address the problem? Why is your approach important and how will your study contribute to the current literature? It seems that one contribution of your study is to add ownership to this line of research. Thus, you should reframe your introduction to highlight ownership as the boundary condition in the mentioned relationships.
  2. It is better to have a part of Literature Review and Hypotheses, though you may not need theoretical argument for each hypothesis. Define the five organizational factors and articulate how they may influence worker outcomes. Explain why ownership could influence the effect of five organizational factors.
  3. Retest your model using moderated regression approach.
  4. In your Discussion part, give some headings, such as theoretical implication, practical implication, limitations and directions for future research. In both theoretical and practical implications, use First, Second, and Third to specify these implications.

Reviewer 3 Report

The authors analysed the safety and health programs in nursing homes via a survey instrument developed in the scope of the Total Workers Health initiative. It was intended to analyse the effect of ownership of the nursing home on the implementation of six different dimensions of safety and health at work. In general, implementation of safety and health measures was relative high. No statistically significant difference could be detected depending on ownership – for profit or non profit. The authors mention correctly that the assessment of ownership might have been to crude and that the power for detecting smaller differences was low.

In general, the study was well conducted and the paper is well written. I have only three minor comments

1) The second sentence of the results section, I would move to the method section

2) page 6 line 234: a point is missing after [21-24]

3) This might be a more important point. The directors of the nursing homes filled out the survey. In the limitation section, you mention that desirability or optimism biases are possible. This is true for all self-administered questionnaires. However, here I think it is more important to discuss whether the survey shows that the directors are well informed about how safety and health at the workplace should be organized or whether the survey depicts the reality in the nursing homes.

Thank you for the opportunity to read this interesting paper.

Round 2

Reviewer 2 Report

I check my comments on the first version of the manuscript. I do not find that you have revised their manuscript according to these comments. For example, the first comment asks you to give some hypotheses and then test them. I provide many examples for your reference. Unfortunately, I do not see any revision to address this "major concern". In the case, I have to suggest your reconsidering these comments in your further revision on the manuscript.

Author Response

Author Response: We apologize for misunderstanding your initial request. We have updated the manuscript to clearly state our hypotheses. We appreciate the list of suggested hypotheses provided. The instrument will allow for a wide range of useful hypotheses to be tested in the future, included some provided in the initial review; we have updated the introduction to note the opportunity for future research to investigate some of these stimulating hypotheses.

Manuscript Updates:

Introduction, lines 43-54: The Workplace Integrated Safety and Health Assessment (WISH) is an organizational level measure of workplace polices, programs, and practices that affect worker safety health and wellbeing. The tool, informed by the Total Worker Health® framework, was developed to allow employers and researchers to assesses how well best practices for supporting work safety, health, and wellbeing are implemented and was recently vali-dated using item response theory analysis. There are five domains in the validated measure: leadership commitment; participation; policies program and practices that foster supportive working conditions; comprehensive and collaborative strategies; and adherence to federal and state regulations and ethical norms. Administration of the WISH will allow for the evaluation of various hypotheses around TWH policies and practices and associated outcomes; for example, do institutions with high scores on the policies program and practices domain have lower rates of worker injury?

Introduction, lines 81-89: The purpose of this study is to determine the degree to which nursing homes implement TWH approaches as measured by the WISH and determine whether their ownership type, for-profit and/or corporate, is associated with their scores, after controlling for other organizational characteristics. We hypothesize that after adjusting for nursing home characteristics (1) for-profit ownership will be associated with lower uptake of THW approaches as measured by the WISH, (2) That corporate ownership will be associated with lower uptake of THW approaches as measured by the WISH. To our knowledge this is the first comprehensive survey in nursing homes of implementation of workplace polices, programs, and practices that can affect worker safety, health, and wellbeing.